# Application of Business Simulation Games in Flipped Classrooms to Facilitate Student Engagement and Higher-Order Thinking Skills for Sustainable Learning Practices

**Ching-Yun Hsu and Ting-Ting Wu ***

Graduate School of Technological and Vocational Education, National Yunlin University of Science and Technology, No. 123, Sec. 3, University Rd., Douliu 64002, Taiwan; d11143002@yuntech.edu.tw
* Correspondence: danytingting@gmail.com

**Abstract:** It is very important to adopt innovative digital technologies in educational systems to overcome the challenges in modern learning environments, especially in the post-COVID-19 era. The fourth Sustainable Development Goal (SDG) of the 2030 Agenda is supported by new educational trends that consider game-based learning as a pedagogical method in the classroom. Teaching sustainability management in higher education institutions with innovative digital tools plays a fundamental role in the transition toward sustainable societies. Suitable game design elements play a significant role in facilitating sustainable learning. This study explored the effectiveness of incorporating business simulation games with project-based learning (PBL) in a flipped classroom setting. This approach was adopted within the context of a university cross-border e-commerce course to prepare students for acquiring 21st-century skills such as higher-order thinking skills in a rapidly changing educational landscape. A quasi-experimental method was employed, involving a total of 60 university students from China's Zhejiang Province. Participants completed an online questionnaire designed to assess their learning engagement across three dimensions (cognitive, emotional, and behavioral) as well as their higher-order thinking skills (problem-solving, critical thinking, and creativity). The results show that the business simulation games combined with flipped classroom learning had a significantly positive impact on students' learning outcomes, enhancing their problem-solving, critical thinking, and creative capabilities. Importantly, this approach also improved student engagement and promoted sustainable practices by applying real-life scenarios in an interactive environment. We conclude that business simulation games integrated with project-based learning (PBL) in flipped classroom settings represent a valuable educational approach. This approach not only enhances learning engagement but also fosters the development of higher-order thinking skills, encouraging students to adopt sustainable learning practices.

**Keywords:** business simulation games; higher-order thinking skills; sustainable learning practices



## 1. Introduction

The COVID-19 pandemic has adversely impacted progress towards achieving the Sustainable Development Goals (SDGs) related to education. These goals strive to provide lifelong opportunities for both youth and adults by equipping them with essential skills and knowledge [1]. In this context, educational institutions bear a crucial responsibility to impart students with the fundamentals required for sustainable learning, thus enabling them to effectively surmount complex and challenging situations. To cope with sudden market changes and thrive in a complicated environment. Business simulation games are instrumental in achieving this goal, because they provide a true representation of market processes in a safe and dynamic virtual environment and can assist in developing professional and decision-making skills while considering real-world surroundings [2–6]. Moreover, the gaming platforms help students put theories with flipped learning into

practice for enhancing engagement and higher-order thinking skills in a virtual environment. Consequently, these institutions must ensure that as learners successfully navigate these challenges, they make a vital contributions towards the fulfillment of the Sustainable Development Goals (SDGs) in their respective nations [7,8]. In the wake of the COVID-19 pandemic, college graduates are facing formidable difficulties in their job searches. However, the disruptions caused by the COVID-19 pandemic have fueled the growth of online entrepreneurship [9], opening up a myriad of new opportunities [10]. Simultaneously, the burgeoning digital economy has led to the emergence of novel entrepreneurial avenues such as cross-border e-commerce, influencer marketing, and game streaming [11], offering a wealth of new entrepreneurial prospects for university graduates. The establishment and openness accessibility of digital platforms have considerably reduced the barriers to entry and the costs traditionally associated with entrepreneurship. This undoubtedly creates an environment conducive to university students with limited financial resources who aspire to venture into the entrepreneurial landscape [9].

Cross-border electronic commerce (CBEC) refers to international trade conducted through the internet and encompasses electronic commerce transactions, payment settlements, and international logistics transportation across different countries [12]. CBEC has become a significant contributor to regions within the global economy driven by e-commerce, rapidly developing to become one of the most resilient platforms [13]. CBEC and entrepreneurship have increasingly gained academic recognition in recent years, leading several higher education institutions in China to embrace related educational programs. However, the rapid development of CBEC has led to most B2C platforms gradually canceling individual registration, and instead only allow the registration of accounts for licensed companies or brands. This shift places a financial obstacle in the way of enterprising students who lack the funds for company registration. As a result, teachers and students alike are unable to engage in sustainable learning practices in B2C CBEC platform activities, with courses comprising lectures and discussions that may provide theoretical insights but fail to provide real-world experiences, leaving students with minimal training on project implementation and practical learning [14,15]. This further decreases students' classroom engagement and limits their opportunities for cultivating higher-order thinking skills and abilities. Several studies have highlighted that the traditional techniques and methods for teaching entrepreneurship and business courses do not adequately prepare learners to adapt to sudden market changes or operate in complex, real-world scenarios [16–18]. To address this issue, computer-assisted learning, including business simulation games, can realistically depict market processes within a secure, sustainable, digital, and dynamic virtual environment. These games play a pivotal role in cultivating professional and informed decision-making while increasing learning engagement and higher-order thinking skills by prompting individuals to make choices similar to those made in real-world scenarios [19,20]. Several studies have advocated for the use of business simulation games to boost creativity, individual motivation, critical thinking, team management, collaborative skills, time management, experiential learning, and dedication to entrepreneurship [21–25].

Based on these advantages, this study aimed to implement a business simulation game, "CEMO Simulation", in combination with project-based learning (PBL) in a flipped classroom. Despite numerous studies exploring the use of business simulation games, there has been limited research on the application of CBEC-based games in flipped classrooms and intermediate courses to enhance learning outcomes through the development of higher-order thinking skills and increased learning engagement. Moreover, the convergence of flipped learning and gamification represents a new teaching methodology and an innovative approach tailored to the evolving demands of education in the new millennium [26,27].

## 2. Literature Review

### 2.1. Educational Considerations Regarding the Flipped Classroom

The flipped classroom describes a mixed methodology that combines face-to-face and virtual teaching methods and is currently being employed at all educational levels. Its increasing use has been attributed to its overall effectiveness and the specific practical components that constitute its training approach [27–29]. However, to maximize the benefits of this innovative pedagogical approach, it is essential to ensure active student engagement to facilitate the effective acquisition of knowledge [30]. There is a close relationship between the flipped classroom and the instructional objectives of Bloom's Taxonomy. In traditional teaching settings, teachers primarily engage in knowledge dissemination. In the flipped classroom, students independently study basic concepts, completing lower-level learning (i.e., memorization and comprehension) by themselves, while higher-level learning skills (i.e., application, analysis, evaluation, and creation) are achieved through interaction with educators in the classroom [31]. Research has found that the flipped classroom not only encourages students to engage in active learning before class, but also improves their class learning [32]. Moreover, educators are tasked with providing engaging activities within the classroom that promote team work through group projects, coordinated discussions, debates, lectures, pitch reports, presentations, gamification, and other forms of active participation [33]; higher-order thinking [34]; and learning achievements [35]. The literature has reported the successful implementation of flipped classrooms across various fields, including engineering [36], mathematics [37], education [38], business [39], and entrepreneurship [40], to name a few. Conventional educational systems have failed to nurture the essential skills students need to apply theoretical knowledge in real-world businesses [41,42]. Furthermore, flipped learning has the potential to address the limitations of traditional, teacher-centered teaching strategies by adopting a student-centered model that motivates students to apply theoretical knowledge and essential skills in practical contexts using a wide range of activities [43–45]. Based on these findings, the present study adopted the flipped classroom as the teaching methodology to increase student engagement and sustainable learning practice.

### 2.2. Particulars of Game-Based Learning in Education

According to Gabrielsson, Tell, and Politis [46], business schools and policymakers have faced substantial criticism for disproportionately burdening students with theoretical and academic knowledge instead of equipping them with practical, real-world skills, a concern that has been shared by both researchers [47] and educators [48,49]. Consequently, there are compelling reasons to adopt new technologies in educational systems. According to Abourezk [50], students tend to acquire a deeper understanding when they engage with experiential and practical knowledge rather than passively listening to classroom lectures. Business simulation games, which provide an interactive, exciting, and enjoyable learning environment [51,52], have been supported by researchers whose work has revealed that business simulation games can increase motivation [53] and enhance creativity and learning [54,55]. Furthermore, business simulation games have gained widespread use in business education to enhance engagement, improve higher-order thinking, and achieve specific learning objectives. Several studies have demonstrated the effectiveness of game-based learning in reinforcing theoretical concepts and creating an immersive learning environment that aids students in developing higher-order thinking skills through challenging problem-solving tasks [2,56,57]. Researchers have highlighted the use of business simulation games in business and entrepreneurial studies, marking a paradigm shift from conventional teaching methods toward innovative teaching practices.

### 2.3. Educational Considerations Regarding Project-Based Learning

Education should adjust to a dynamic world, and project-based learning (PBL) is gaining popularity as it effectively addresses this demand [58,59]. PBL is an educational model centered on project-based activities [60]. In the PBL process, students identify

problems, develop the skills to gather and integrate information, enhance communication through group discussions, and work collaboratively to propose solutions [61,62]. In higher education, PBL equips students with a varied spectrum of knowledge and essential innovative skills, enabling them to effectively navigate future challenges and achieve success [63]. PBL has the potential to positively influence students' attitudes toward learning, leading to increased positive effects on both student learning effectiveness and engagement [63–66]. PBL is an inquiry-based, holistic instructional approach grounded in authentic contexts. It embodies a unique form of collaborative learning that prioritizes student-centered engagement with concrete, real-world artifacts [67,68]. Project-based learning (PBL) closely resembles real-world business scenarios and has been extensively adopted in higher education, particularly in the field of business education for authentic projects with actual corporations. The research presented in [65] implemented PBL in a business informatics university course. In this scenario, students acquire knowledge through practical application during the project elaboration, aligning with the principles of PBL. Throughout the process, students participate in hands-on activities, including exploring the basics of data processing, conducting data analysis, modeling business processes, and developing a simple system. Additionally, this pedagogical approach proves to be a highly effective method that seamlessly integrates into dynamic and demanding learning environments such as international business education [64]. In higher education, PBL enables students to gain a broad spectrum of knowledge and essential innovative skills crucial for addressing future challenges and attaining success [63]. Several studies have highlighted the positive influence of PBL on students' attitudes toward learning, resulting in enhanced effectiveness and engagement [63–66]. Using PBL to promote teacher–student interaction and foster students' active problem-solving skills in flipped classrooms necessitates the cultivation of higher-order thinking skills.

Numerous researchers have explored how the PBL approach can lead to higher-order thinking skills. These studies investigated PBL's impact on students' higher-order thinking skills via follow up questionnaires, and its effectiveness was confirmed by the empirical data. For instance, Sulisworo [36] attempted to enhance higher-order thinking skills through a PBL approach to STEM education by arranging a study in which the experimental group followed a PBL approach while the control group used traditional scientific learning methods. The results indicated a positive impact of the PBL approach on the students' higher-order thinking skills.

### 2.4. Learning Outcomes of Student Engagement

"Student engagement" refers to the effort that a student invests in learning activities and is influenced by a variety of factors and classroom dynamics [69]. Regardless of the mode of learning, student engagement has been a focus of several studies due to its strong correlation with learning outcomes [70]. Engagement constitutes a critical component in any educational endeavor, encompassing three key facets: behavioral, cognitive, and emotional engagement. Although student engagement is crucial, there remains a gap in the existing literature concerning engagement within the context of PBL in flipped classrooms, particularly when integrated with business simulation games for students majoring in CBEC-related courses. Therefore, this study sought to bridge this gap by providing insights on student engagement in business school settings that combine PBL and flipped classrooms where business simulation games for CBEC practice have been incorporated. Our findings can aid educators in developing a more engaging curriculum and learning activities that promote active learning and advanced skills.

### 2.5. Learning Outcomes of Higher-Order Thinking Skills (HOTS)

Recently, universities have been promoting higher-order thinking skills (HOTS) by incorporating technology into various educational contexts [70]. Attending a university offers students additional advantages, including the opportunity to nurture creative abilities that will not only serve them well in the workplace, but also in society. Problem-solving,

a key component of HOTS, requires the capacity to identify issues, collect and analyze relevant facts to generate solutions, and then, take decisive action. Creativity in this context means the ability to examine information objectively, think rationally and logically, and reach reasonable conclusions [70–72]. Hwang et al. [73] defined creativity as the ability to come up with new ideas, innovative concepts, and alternative approaches by explaining, modifying, exploring, and evaluating existing knowledge and resources. Kim et al. [74] examined the effectiveness of measuring educational success using HOTS and explored how it could be implemented in this context. Furthermore, in conjunction with continuous and rapid technological advancement, it is necessary that university students acquire new competencies in order to thrive in society. These competencies, often referred to as 21st-century skills, encompass high-level cognitive abilities. Therefore, HOTS incorporate 21st-century skills, encompassing the abilities students will need to succeed in the future [75].

Higher-order thinking and metacognitive skills are cultivated through engagement in classroom activities that motivate students to actively participate. Some studies have found that learning ability and engagement are closely associated with a student's HOTS [76]; therefore, educators need to explore and develop teaching methods that will nurture these skills [77]. The significance of HOTS is also evident in global skill studies that emphasize problem-solving skills and creativity in diverse and unanticipated environments [78], suggesting that HOTS is a vital indicator of the effectiveness of classroom instruction and employability [3].

In light of the existing literature, this study aimed to fill the present gap in research relating to business schools' focus on student engagement and higher-order thinking skills in the context of PBL in flipped classrooms integrating business simulation games. To evaluate the effectiveness of the proposed learning method. An experiment was con-ducted to answer the following research questions:

1. Can business simulation games (BSGs) enhance students' behavioral engagement in comparison with flipped classroom learning (FCL)?
2. Can business simulation games (BSGs) enhance students' emotional engagement in comparison with flipped classroom learning (FCL)?
3. Can business simulation games (BSGs) enhance students' cognitive engagement in comparison with flipped classroom learning (FCL)?
4. Can business simulation games (BSGs) promote students' problem-solving in comparison with flipped classroom learning (FCL)?
5. Can business simulation games (BSGs) promote students' critical thinking in comparison with flipped classroom learning (FCL)?
6. Can business simulation games (BSGs) promote students' creativity in comparison with flipped classroom learning (FCL)?

### 3. Methodology and Data Collection

#### 3.1. Overview of Research Design

This study employed a mixed-method quasi-experimental design that integrated both quantitative and qualitative analyses to gain a comprehensive understanding of the phenomenon under investigation. A key feature of quasi-experimental designs lies in their ability to address a special aspect of the experimental group while simultaneously keeping all other elements constant between the experimental and control groups. The quasi-experimental design utilizes two types of variables: (1) independent variables, characterized by two different instructional strategies (flipped classroom learning and flipped classroom learning with simulation game learning), and (2) dependent variables, assessed through an evaluation of students' engagement and higher-order thinking skills. The research design for group comparison research is shown in Figure 1.

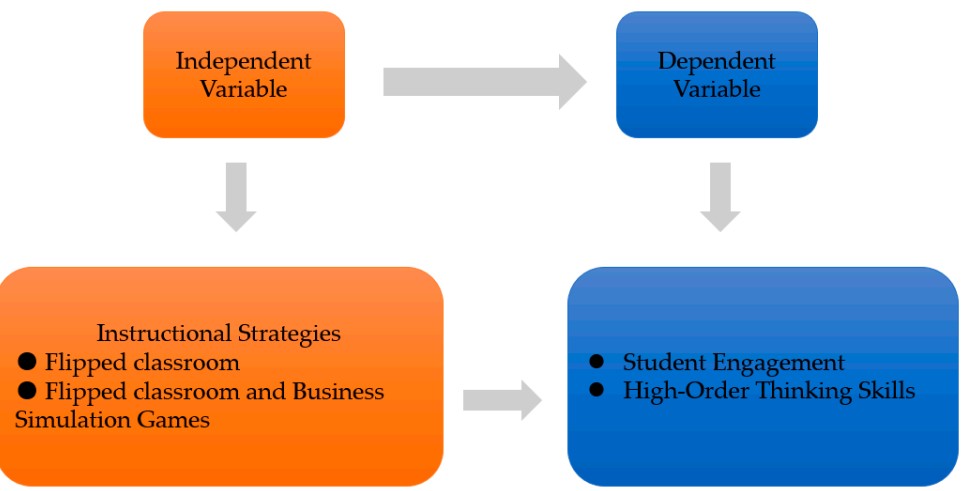

**Figure 1.** Research design.

*3.2. Research Process*

3.2.1. Experimental Procedure

The experiment consisted of nine class meetings, with each class lasting 80 min. An overview of the experimental procedure is presented in Figure 2. The initial period involved the distribution of pre-test questionnaires and a brief course lecture. After introducing the fundamental concepts, students in the experiment group were introduced to business simulation games for learning. All tasks required to complete the games were thoroughly explained, and the students were divided into groups of six members each to engage in various projects and tasks. From the first to the sixth meetings, students in the experimental group studied CBEC management using the PBL in flipped classrooms and business simulation games. Meanwhile, students in the control group also used PBL with flipped classroom learning, sans BSGs. Data collection occurred both at the outset and at the conclusion of the experiment. Before administering the questionnaires to the participants, researchers explained the research study's objectives. Following data collection and analysis, the researchers used the results to select participants for the interviews. During the seventh class, each team presented CBEC business plans, or "learning outcomes", which were evaluated by their instructor. In the eighth class, students completed post-test questionnaires. In the final stage (Class 9), ten students from the experimental cohort took part in interviews, with each interview lasting 40 to 70 min. The interview questions were designed to elicit additional insights into the experimental methods and to complement the quantitative data.

3.2.2. Experimental and Control Group HOTS Activities

Both the experimental and control groups were instructed by the same teacher, and the teaching content in each class was identical. Table 1 presents a detailed overview of the experimental and control group activities related to HOTS. The course emphasized the establishment of an effective virtual CBEC marketing and operating environment, encompassing market environments, the data analysis of eCommerce platforms, as well as specific operations and decision-making. The course content was organized into six sections. The first section involved a brief lecture introducing the concepts of business generation and market opportunity analysis. Section 2 provided details about procurement and supply chain management. Section 3 focused on product innovation within the context of a business model. Section 4 involved designing a marketing strategy within a business model framework. Section 5 largely consisted of financial analysis, while a comprehensive business strategy was developed in Section 6.

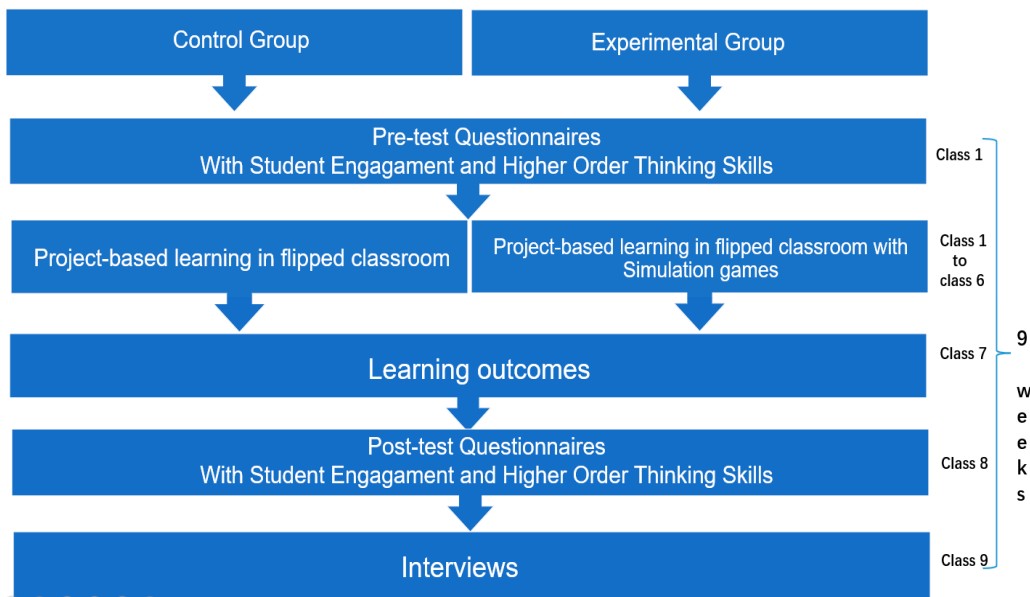

**Figure 2.** The experimental procedures of this study.

**Table 1.** The experimental and control group HOTS activities.

| No (Section) | Instructional Content with PBL | Group | | HOTS Activities | | |
|---|---|---|---|---|---|---|
| | | Control Group | Experimental Group (CEMO Simulation) | Problem-Solving | Critical Thinking | Creativity |
| 1 | Introduction<br>• Business introduction<br>• Market opportunity analysis<br>• Create a new company | Video assignment: how to create a new business/company<br>• Confirm company mission<br>• Conduct a market analysis<br>• Determine an appropriate new company | | • Analyze market information and market demand<br>• Create new company | • Determine how to operate company | • Design a company name and logo |
| 2 | Business process<br>• Procurement<br>• Supply chain management<br>• B2C wholesale | Video and slideshow: procurement management<br>• Customize initial demand<br>• Perform material requirement planning<br>• Determine purchase price<br>• Implement supply chain management | | • Determine and analyze the market value<br>• Develop a procurement plan | • Assess supply chain competition and company sales | • Integrate idea generation with company plan and forecast future market behavior |
| 3 | Business model<br>• Create a new shop on the e-commerce platform<br>• Create posts to sell products via e-commerce | Video assignments: how to register a new company and upload products<br>• Register new account<br>• Obtain required materials for product listing and product details to upload to the platform<br>• Determine the sale price for each item | | • Determine and analyze the production plan and create product listings | • Determine the best way to upload product | • Design innovative products |
| 4 | Marketing strategy<br>• Marketing programs<br>• Social media marketing<br>• Marketing fees | Video and slideshow: how to analyze marketing programs<br>• Assess the competitive landscape within the market<br>• Conduct a marketing program for the company<br>• Develop social media marketing | | • Analyze marketing program<br>• Develop social media marketing | • Analyze data from marketing program<br>• Conduct innovative marketing strategies to increase sales growth | • Design creative social media and marketing activities |

**Table 1.** *Cont.*

| No (Section) | Instructional Content with PBL | Group | | HOTS Activities | | |
|---|---|---|---|---|---|---|
| | | Control Group | Experimental Group (CEMO Simulation) | Problem-Solving | Critical Thinking | Creativity |
| 5 | Business plan<br>• Assess initial costs<br>• Determine loan parameters<br>• Delinquency payouts<br>• Financial analysis | Slideshow: how to create a financial report<br>• Request financial aid<br>• Analyze and create reports<br>• Analyze profit reports<br>• Analyze inventory data reports | | • Conduct financial analysis<br>• Conduct order report, profit report, and inventory report | • Evaluate financial status<br>• Determine optimization scheme to improve cost structure<br>• Reduce inventory costs and enhance inventory turnover | • Design innovative solutions for cost reduction<br>• Explore optimization of financial operations |
| 6 | Business Strategy<br>• Create a company model<br>• Create a business plan display | Video and slideshow: business model<br>• Create a company model<br>• Present business plan | | • Analyze business strategies and make necessary adjustments<br>• Develop business model | • Formulate a financial strategy | • Design a business plan |

### 3.3. Business Simulation Games (BSGs)

All the PBL tasks required to complete the games were discussed with the students through groups. This research utilized a simulation game licensed under the name "Cross-border e-Commerce Marketing and Operating Simulation" (CEMO Simulation) and was specially designed to provide a virtual CBEC operating environment. The online BSG CEMO Simulation encompasses market environments, data analysis of eCommerce platforms, and other specific operations. It offers further flexibility through the integration of various business aspects during simulations to regulate the difficulty level of the player experience. We applied the PBL in a flipped classroom using CEMO Simulation to enhance higher-order thinking skills in sustainable learning practices. Each CEMO Simulation section within the games included a brief description and learning activity designed according to the PBL approach, as detailed below:

1. Introduction: In this section, students begin the game by starting a new company. Students assume various roles within the company and take responsibility for their positions in order to manage the firm. This section covers a business introduction, CBEC market opportunity survey, and market environment analysis.
2. Procurement: This section involves activities related to material requirement planning and supply chain management decisions. The game gradually introduces new decision-making content relevant to the company's procurement life cycle.
3. Production: Within this section, students select a B2C CBEC platform and create posts to buy and sell. Students need to present their decisions in a logical sequence.
4. Marketing: This section focuses on marketing programs and marketing strategy. Students are tasked with reinforcing their strategic decisions by linking them with cash flow and profitability considerations.
5. Finance: The final section allows the instructor to customize the initial costs, loan parameters, and funds. Students need to make repeated decisions regarding essential loans, costs, and profits, as well as develop strategies to maximize profitability.

Students were given the opportunity to create an e-commerce shop, identify goals, evaluate market surveys, develop strategies, upload innovative products to target specific markets, and develop new and innovative products (see Figure 3). The games are organized into sections based on PBL activities to encourage students to make strategic decisions according to potential consequences. In flipped classrooms, it is essential that students

grasp the theoretical foundation of the material prior to class; once in the class, the instructor provides additional details about the overarching framework and guides the students as they complete the game tasks. Students' primary objectives with the simulation games are to identify business challenges and find ways to address them. When the games end, CEMO Simulation provides the instructor with a report regarding the students' development, along with additional information. The instructor can then guide each group of students in summarizing their decision-making processes, identifying problems encountered in the games, and analyzing the reasons for a company's success or failure, including instances of bankruptcy. Each group shares their thoughts and engages in discussions and reflection sessions, helping students connect their game experiences to the real world, promoting the learning process. At this stage, the most innovative group is selected and recognized for having gained practical experience. Finally, the instructor provides feedback on the simulation games and each group's reflections using evaluations.

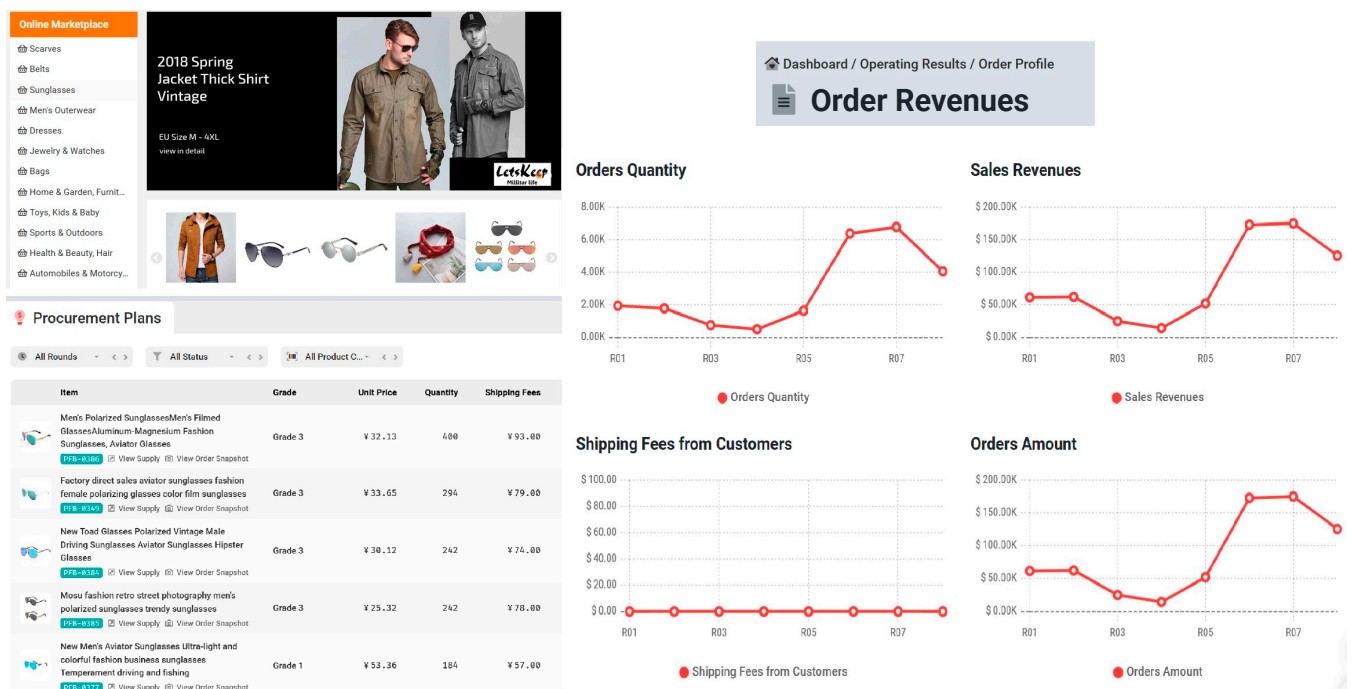

**Figure 3.** The CEMO Simulation interface.

### 3.4. Participants

This study included 60 participants, all of whom were sophomore students majoring in international e-commerce at a university in Zhejiang Province, China. These students were divided into two groups: the experimental group (*n* = 30) and the control group (*n* = 30).

### 3.5. Research Instruments

This research implemented a pre-test, a post-test questionnaire, and interviews conducted throughout the experiment. The questionnaire on students' engagement was developed based on Reeve and Tseng [78] and consisted of 17 items organized into three domains: behavioral, cognitive, and emotional engagement. The reliability test results are presented in Table 2, where Cronbach's alpha values are 0.894, 0.865, and 0.918 for the behavioral, emotional, and cognitive domains, respectively. As all values are higher than the suggested threshold value of 0.7, the results can be considered reliable. The HOTS questionnaire was adopted from a study by Hwang et al. [70], which consisted of 11 items organized into three domains: problem-solving, critical thinking, and creativity. Problem-solving involves identifying problems and analyzing relevant materials and in-

formation. Critical thinking pertains to the cognitive ability to make logical judgments. Finally, creativity refers to the capacity to create and improve on original concepts. Table 2 shows that all constructs related to higher-order thinking skills were considered reliable, with Cronbach's alpha values of 0.925, 0.940, and 0.768, respectively. Again, good reliability is determined by values exceeding the suggested threshold value of 0.7.

**Table 2.** The instrument reliability test results.

| Variables | Construct | Reliability |
|---|---|---|
| Student engagement | Behavioral engagement | 0.894 |
| | Emotional engagement | 0.865 |
| | Cognitive engagement | 0.918 |
| Higher-order thinking skills | Problem-solving | 0.925 |
| | Creativity | 0.940 |
| | Creativity | 0.768 |

*3.6. Data Collection*

Data were collected via pre- and post-test questionnaires that measured classroom engagement and higher-order thinking among participants in both the experimental and control groups. These questionnaires were then collected and the quantitative data were entered into SPSS 25.0 for data analysis. For statistical analysis, descriptive statistics and a one-way analysis of covariance (ANCOVA) were used. The results indicated homogeneity in behavioral engagement (F = 0.081, *p* = 0.777), emotional engagement (F = 0.287, *p* = 0.594), and cognitive engagement (F = 0.853, *p* = 0.360), as well as problem-solving (F = 0.377, *p* = 0.542), critical thinking (F = 3.18, *p* = 0.080), and creativity (F = 0.881, *p* = 0.352). These results confirmed that the regression coefficients of the groups did not reach significant levels, supporting the assumption of homogeneity in covariance analysis and allowing for the use of one-way ANCOVA to test for significant differences between the groups' pre- and post-test scores. Additionally, interviews were conducted with students to gain insights into their learning experiences. The interview data were recorded and transcribed, and comprehensive details of these interviews are provided in Appendix A.

## 4. Analysis and Results

*4.1. Analysis of Student Engagement*

In line with the proposed questions and their underlying framework, the complete and usable pre- and post-test data obtained from the 60 participants in the quasi-experiment were used for the dependent variables. Descriptive statistics were used to analyze these variables (as shown in Table 3), revealing improvements in student engagement for both the experimental and control groups following their participation in the training courses. A subsequent analysis of covariance (ANCOVA) test was used to assess differences between the pre- and post-test scores for all dependent variables. The results indicated significant differences in the post-test scores for behavioral engagement and cognitive engagement between the two groups, as shown in Table 4.

**Table 3.** The descriptive results for student engagement.

| Variable | Group | Pre-Test | | Post-Test | |
|---|---|---|---|---|---|
| | | **M** | **SD** | **M** | **SD** |
| Behavioral engagement | Experimental | 3.48 | 2.111 | 4.69 | 1.478 |
| | Control | 3.07 | 0.922 | 4.43 | 2.520 |
| Emotional engagement | Experimental | 3.21 | 1.507 | 4.59 | 1.671 |
| | Control | 2.95 | 1.006 | 4.39 | 2.434 |
| Cognitive engagement | Experimental | 3.17 | 2.102 | 4.61 | 2.149 |
| | Control | 2.95 | 1.422 | 4.26 | 3.960 |

**Table 4.** The ANCOVA results for student engagement.

| Variable | SS | df | Mean Square | F | *p* | Partial η² |
|---|---|---|---|---|---|---|
| Behavioral engagement | 0.911 | 1 | 0.911 | 5.274 | 0.025 * | 0.085 |
| Emotional engagement | 0.446 | 1 | 0.446 | 2.518 | 0.118 | 0.042 |
| Cognitive engagement | 1.382 | 1 | 1.382 | 6.581 | 0.013 * | 0.104 |

\* $p < 0.05$.

### 4.1.1. Analysis of Behavioral Engagement

In terms of behavioral engagement, the ANCOVA results in Table 4 show significant differences between the groups (F = 5.274, $p < 0.05$, 0.025). Based on the post-test mean scores (see Table 3), students in the experimental group (M = 4.69, SD = 1.478) showed higher levels of behavioral engagement compared to students in the control group (M = 4.43, SD = 2.520). These results support the assertion that students in the experimental group demonstrated significantly higher behavioral engagement after participating in the BSG learning activities, thereby confirming research question 1.

### 4.1.2. Analysis of Emotional Engagement

Regarding emotional engagement, the ANCOVA results in Table 4 show no significant differences between the groups (F = 2.518, $p = 0.118$), resulting in the rejection of research question 2. However, it can still be observed from Table 3 that the post-test mean score for students in the experimental group (M = 4.59, SD = 1.671) was higher than that of the students in the control group (M = 4. 39, SD = 2.434).

### 4.2. *Analysis of Higher-Order Thinking Skills*

Table 5 provides descriptive statistics showing that both the experimental and control groups demonstrated improvements in higher-order thinking skills following the course training. Subsequently, an analysis of covariance (ANCOVA) was used to analyze the differences between the students' pre- and post-test scores. The results indicate that there were significant differences in problem-solving, critical thinking, and creativity between the two study groups, as shown in Table 6.

**Table 5.** The descriptive results for HOTS.

| Variable | Group | Pre-Test | | Post-Test | |
|---|---|---|---|---|---|
| | | M | SD | M | SD |
| Problem-solving | Experimental | 3.33 | 1.539 | 4.39 | 2.176 |
| | Control | 2.96 | 1.085 | 4.33 | 1.968 |
| Critical thinking | Experimental | 3.10 | 1.567 | 4.42 | 2.354 |
| | Control | 2.84 | 1.159 | 4.22 | 2.501 |
| Creativity | Experimental | 3.14 | 1.040 | 4.44 | 1.688 |
| | Control | 2.87 | 1.192 | 4.21 | 1.752 |

**Table 6.** The ANCOVA results for HOTS.

| Variable | SS | df | Mean Square | F | *p* | Partial η² |
|---|---|---|---|---|---|---|
| Problem-solving | 31.533 | 1 | 31.533 | 18.563 | 0.000 *** | 0.249 |
| Critical thinking | 11.554 | 1 | 11.554 | 6.620 | 0.013 * | 0.106 |
| Creativity | 8.422 | 1 | 8.422 | 6.790 | 0.012 * | 0.108 |

\* $p < 0.05$, \*\*\* $p < 0.001$.

### 4.2.1. Analysis of Problem-Solving

The ANCOVA results in Table 6 show significant differences in problem-solving scores between the two groups (F = 18.563, $p < 0.05$, 0.000). As can be seen in Table 5,

the experimental group had significantly higher average post problem-solving scores (M = 4.39, SD = 2.176) than the control group (M = 4.33, SD = 1.968). These results indicate that the use of BSG learning in a flipped classroom led to greater improvements in students' problem-solving skills compared to a flipped classroom, thereby confirming question 4.

### 4.2.2. Analysis of Critical Thinking

Regarding critical thinking, the ANCOVA results in Table 6 show significant differences between the two groups (F = 6.620, *p* < 0.05, 0.013). Based on the post-test mean scores (Table 5), students in the experimental group had higher scores (M = 4.42, SD = 2.354) than the control group students (M = 4.22, SD = 2.501). This indicates that the experimental group demonstrated significantly higher critical thinking abilities compared to the control group, thereby confirming question 5.

### 4.2.3. Analysis of Creativity

With respect to creativity, the ANCOVA results in Table 6 also show significant differences between groups (F = 6.790, *p* < 0.05, 0.012). As seen in Table 5, the experimental group had higher post-test mean scores (M = 4.44, SD = 1.688) compared to the control group (M = 4.21, SD = 1.752). These results indicate that the experimental group exhibited significantly higher creativity compared to the control group, which can be attributed to the implementation of the BSG intervention, thereby confirming question 6.

## 5. Discussion

This study assessed participants' behavioral engagement, emotional engagement, cognitive engagement, and higher-order thinking skills following the integration of business simulation games using a project-based learning approach in a flipped classroom. The results of this study support the use of BSGs to enhance behavioral and cognitive engagement while cultivating HOTS to promote sustainable learning and practices. In the PBL flipped classroom, students enrolled in CBEC courses following a course schedule arranged by their instructor while combining strategies including weekly previewing and reporting. This approach promotes active learning and leverages the principles of PBL to enhance teamwork, and encourage the application of knowledge for solving tangible problems [61,62,79]. Incorporating BSGs into this context creates a learner-centered environment that can significantly improve students' cognitive and behavioral engagement through systematic operations while contributing to the development of their higher-order thinking skills [36]. The results show a beneficial impact on student engagement and higher-order thinking skills compared to previous studies, which is in line with the previous findings [2,30,55]. In addition, the results show that involving BSGs enables the learning of skills by simulating real-time experiences in the virtual environment, which is consistent with earlier findings [30,80] and also in line with a study conducted by Deterding et al. [80].

The apparent lack of effect on emotional engagement may be attributable to the following considerations: According to interviews with students, the course utilized in this experiment is a mandatory sophomore course and required professional English skills to upload products to the B2C platform and reply to customers with good service. In this course, students need to possess a comprehensive interdisciplinary knowledge base, coupled with a background in financial management to safeguard the company against insolvency, a process that entails a heavy course load and high stress [2]. Within this context, students in both the experimental and control groups must focus on learning related to specific course tasks, directly impacting their level of emotional engagement during the learning process.

BSGs provide a dynamic and realistic business environment that not only fosters active learning but also establishes a robust collaborative relationship between BSG- and PBL-based flipped classroom activities. This combination of the teaching methodology of BSGs with project-based learning in a flipped classroom setting has been shown to motivate students to become actively engaged in the learning process. Through challenging the

students with tasks that engage them in the learning process, students will constantly engage in solving complex problems and decision-making. These activities foster competition among students through teamwork and cultivate higher-order thinking skills. It is important to emphasize the importance of incorporating new technology of games into the classroom because it enhances engagement, and BSGs should also be included in business and management courses to enable students to explore business operation based on realistic experiences, developing their decision-making skills. To ensure high-quality education, BSGs that support education must be implemented. Thus, with the aforementioned interventions, the promotion of the fourth Sustainable Development Goal (SDG 4) of the 2030 Agenda can be achieved. This objective underscores the importance of providing access to high-quality education for all students and fostering opportunities for lifelong sustainable learning [81]. Moreover, the competitive element in team-based simulations has also been shown to enhance creativity [4]. Teachers also have a key role as they guide students to summarize and review their decisions, identify problems encountered during game-based learning [4], and analyze the reasons for a company's success or failure. Thus, teachers should emphasize enhancing students' behavioral engagement, achievable through class discussions, collaborative group work, and various other activities. Consequently, this approach can elevate the overall level of learning engagement, facilitating the improved development of students' higher-order thinking skills, which is in line with previous findings [2,4].

As each group shares their ideas and leads discussions, these skills are further enhanced. In the final stage, the most creative group is identified and recognized for their contribution to sustainable learning through the practical experience they have obtained, thus encouraging other students to cultivate sustainable learning practices as well.

## 6. Conclusions

The use of BSGs is quickly gaining momentum throughout universities in mainland China, with substantial investments being made to purchase related software. However, in the field of CBEC, many gamified learning approaches merely involve hands-on operations and repetitive actions without actively developing students' higher-order thinking skills. Therefore, the teaching methods employed by instructors play a crucial role in the success of business simulations game learning. In this study, we compared PBL in a flipped classroom setting and PBL to a flipped classroom with the integration of BSGs. The main contributions, limitations, and directions for future research are summarized below.

This study offers several innovative contributions: (1) the combination of BSGs with PBL in flipped classroom learning activities to effectively enhance student engagement and higher-order thinking for sustainable learning practices; (2) the adoption of a quasi-experimental design to allow for an analysis of the differences in student engagement and higher-order thinking between PBL in a flipped classroom and a flipped classroom that incorporate BSGs; and (3) qualitative case interviews to provide a deeper understanding of the factors affecting student engagement and higher-order thinking. (4) For interdisciplinary professional courses, teachers should consider students' curriculum pressure.

Despite these contributions, the limitation of this study was also identified, leading to potential avenues for future research. The study's sample size was relatively small. In the future, more classes and participants should be involved to increase the sample size. Similar courses in the future should explore the relationship of curriculum loading with student engagement and higher-order thinking. Finally, more research on BSG courses could help establish best practices and provide educators with sustainable approaches for implementing BSGs with project-based learning (PBL) in a flipped classroom setting. It is concluded that educational interventions such as BSGs are crucial innovative tools, enabling educators to design a more engaging curriculum and learning activities that promote both student engagement and higher-order thinking skills. Thanks to technological advancements fostering students' holistic development and delivering high-quality education, in accordance with the fourth Sustainable Development Goal (SDG) of the 2030

Agenda [81], there is emphasis on achieving significantly more engaging future education using novel tools and the importance of quality education and skills development [5,81,82].

**Author Contributions:** Methodology, conceptualization, investigation, formal analysis, C.-Y.H.; validation, T.-T.W.; original draft preparation, T.-T.W. and C.-Y.H.; review and editing, C.-Y.H. and T.-T.W.; supervision, T.-T.W. All authors have read and agreed to the published version of the manuscript.

**Funding:** This research is partially supported by the Ministry of Science and Technology, Taiwan, R.O.C., under Grant Nos. MOST 110-2511-H-224-003-MY3 and MOST 111-2628-H-224-001-MY3.

**Institutional Review Board Statement:** Not applicable.

**Informed Consent Statement:** Informed consent was obtained from all subjects involved in this study.

**Data Availability Statement:** The data presented in this study are available on request from the corresponding author. The data are not publicly available due to ethical reasons.

**Acknowledgments:** The authors are grateful to the anonymous reviewers for their insightful comments and suggestions.

**Conflicts of Interest:** The authors declare no conflict of interest. The funders had no role in the design of the study; in the collection, analyses, or interpretation of the data; in the writing of the article; or in the decision to publish the results.

**Appendix A**

Extracts from interviews with students: About behavioral engagement:

1. This kind of classroom operation task is more interesting, and I will be more focused.
2. In the past, classroom teaching only emphasized theory and simple explanations, which were relatively superficial. Nowadays, combined with specific practical operations, teaching can be better implemented.
3. The previous classes were quite dull, with only textbook content explanations, and it was easy to zone out. This type of class-room that combines theory with practical operations is more likely to stimulate students' interest.

About emotional engagement:

1. The teacher's open classroom has inspired our ability to think independently after the game was done, which is very interesting.
2. I required professional English skills to upload products and reply customer with good service.
3. The course workload is heavy and the pressure is relatively high.
4. It requires a background in financial management, avoiding company bankrupt is quite challenging.

About cognitive engagement:

1. When there is a mistake during the operation process, I check where the problem is and correct it. After completing it, I feel a sense of accomplishment in my heart.
2. I use the knowledge I have mastered to create questions and provide answers.
3. If I encounter a question that I don't know how to answer, I search for materials and analyze similar questions to solve it.
4. I reflect on whether my operation is correct.
5. I actively think about the content, significance, and application of learning, and recall relevant information that I have learned before when I am thinking.

About problem-solving:

1. Following the steps to solve problems, if I encounter questions that I cannot solve, I discuss them with team members to find solutions. This approach has enabled me to solve platform questions.

2. One of the important things I have learned is to reflect on and answer the knowledge points after completing the lessons, which has improved my problem-solving ability.
3. When I encounter questions that I don't know how to answer, I either refer to textbooks or search for relevant information online or discuss with my team.

About critical thinking:

1. This teaching method is different from the traditional one, it is designed to work together with the computer to better apply practical concepts.
2. Following the operation process brings new insights, checking to see if the correct steps are completed and refining your own steps. After completing it, there is a great sense of accomplishment.
3. For the questions in the section slides presented by each group of students, I can objectively analyze the rationality, logic, relevance, etc., of the question.
4. Through this course, I have gradually learned to use creativity to view both problems and myself.
5. After independently completing a problem and getting the correct answer, there is a great sense of satisfaction in my heart.
6. During the operation process, I reflect on whether the operation is correct and check to ensure correct completion. After completing it, I have a great sense of achievement and satisfaction.

About creativity:

1. I try to work on new problems and strive to complete them independently.
2. I attempt to challenge new tasks without any guidance and try to complete them independently.
3. I constantly try to work on new problems and independently solve them or complete the tasks.

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
