# Peer review of "Application of Business Simulation Games in Flipped Classrooms to Facilitate Student Engagement and Higher-Order Thinking Skills for Sustainable Learning Practices"

_sustainability, doi:10.3390/su152416867_

Round 1
Reviewer 1 Report
Comments and Suggestions for Authors
Dear authors, please find attached the revisions

Author Response
We feel great thanks reviewer work in our article. According reviewer comments, we laid out problems below with number and response in italics font. Meanwhile, we changes/ additions listed below and the manuscript changed are also given in the red text.
- Lines10-11 The first sentence is a bit lengthy and complex. Consider breaking it into two simpler sentences for clarity
Response: Thanks to this suggestion, according to the reviewer comments, we have made modifications with simpler sentences.
Abstract: It is very important to adopt innovative digital technologies in the educational systems to overcome the challenges in modern learning environment, especially in the post-COVID-19 era. The 4th Sustainable Development Goal (SDG) of the 2030 Agenda is supported by new educational trends that consider game-based learning as a pedagogical method in the classroom. Teaching sustainability management in higher education institutions with innovative digital tools plays a fundamental role in the transition toward sustainable societies.
- Lines 38-43: While you mention the challenges faced by educational institutions and students, consider being more specific about how these challenges directly relate to your study's focus on business simulation games and flipped classrooms
Response: Thank you for your comments. We have added sentences to clarify how these directly relate to business simulation games and flipped classrooms in red words.
To cope with sudden market changes and thrive in a complicated environment. Business simulation games are instrumental in achieving this goal, because it is a true representation of market processes in a safe and dynamic virtual environment and can assist in developing professional and decision-making skills while considering real-world surroundings [50,66,68,79,80]. Moreover, the gaming platforms help students put theories with flipped learning into practice for enhancing engagement and higher-order thinking skills in a virtual environment.
- Lines140-144: For instance, the sections on flipped classrooms and game-based learning are quite detailed, while the section on PBL could benefit from additional detail or examples to align with the depth of other sections.
Response: Thank you for suggestions and we added one paragraph in PBL section accordingly the literature.
Education should adjust to a dynamic world, and project-based learning (PBL) is gaining popularity as it effectively addresses this demand. [69,70] PBL learning is an educational model centered on project-based activities [53]. In the PBL learning process, students identify problems, develop the skills to gather and integrate information, enhance communication through group discussions, and work collaboratively to propose solutions [54, 55]. In higher education, PBL learning based learning equips students with a varied spectrum of knowledge and essential innovative skills, enabling them to effectively navigate future challenges and achieve success [71]. PBL learning is an inquiry-based, holistic instructional approach grounded in authentic contexts. It embodies a unique form of collaborative learning that prioritizes student-centered engagement with concrete, real world artifacts. [72,73]. Project-based learning (PBL) closely resembles real-world business scenarios and has been extensively adopted in higher education, particularly in the field of business education for authentic projects with actual corporations. Additionally, this pedagogical approach proves to be a highly effective method that seamlessly integrates into dynamic and demanding learning environments such as international business education. [74]. Using PBL to promote teacher student interaction and foster students' active problem-solving skills in flipped classrooms necessitates the cultivation of higher-order thinking skills.
- The experimental procedure (lines226-245) is well-detailed, giving a clear picture of the process. However, consider briefly explaining why nine class meetings were chosen and how this duration was sufficient for the study.
Response: Thank you for your comments. In the past, the course and study could be completed in nine classes and nine class meetings accordingly. For the duration, prior to the upcoming experiment, we were conducted preliminary test. It is essential to ensure that students can achieve the expected learning outcomes throughout the learning process before proceeding with the formal experiment. I have added some sentence to make the study clear in the manuscript.
3.2.1. Experimental Procedure
The course focused on the topic how to create a new cross border e-commerce company and business management skills in CBEC filed. Prior to the formal experiment, we were conducted preliminary test in previous class in order to ensure that students can achieve expected learning outcomes throughout the learning process. Accordingly prior pedagogical implementation with game-based learning, the experiment could be completed in nine class meetings. Accordingly, we designed and nine class meetings, each class lasting 80 minutes. An overview of the experimental procedure is presented in Figure 2.
- Lines 351-356: The structural equation model used in this study(https://doi.org/10.1186/s12917-023-03695-0) offers an approach to analyzing relationships which could be informative for interpreting your findings.
Response: Thank you offer the valuable suggestion on the analysis by using the tool structural equation model. According the article of “Boomsma A. (1982). The robustness of LISREL against small sample sizes in factor analysis models. In H. Wold & K JOreskog (Eds.), Systerns underindirect observation (pp. 149-173). New York: Elsevier North-Holland.”. The sample size used in structural equation model typically requires a minimum of 100 or more. However, the samples for our research samples were relatively small so we utilize ANCOVA to analyze. In the future investigations, we’ll collect more samples size and analyze in accordance with review’s valuable suggestions.
- Line370-381: There is a repetition in the section about emotional engagement analysis (lines 370-381). This should be corrected to avoid redundancy and ensure clarity.
Response: We were really sorry for our careless mistakes. We have deleted the repetition and modified it as 4.1.3 Analysis of Cognitive Engagement.,
4.1.3. Analysis of Emotional Engagement
Regarding emotional engagement, the ANCOVA results in Table 4 show no significant differences between the groups (F = 2.518, p = 0.118), resulting in the rejection of hypothesis H2. However, it can still be observed from Table 3 that the post-test mean score for students in the experimental group (M = 4.59, SD = 1.671) was higher than that of the students in the control group (M = 4. 39, SD = 2.434).
4.1.3 Analysis of Cognitive Engagement
The ANCOVA results in Table 4 show significant differences in cognitive engagement between the groups (F = 6.581, p < 0.05, 0.013). As reflected in the post-test mean scores shown in Table3, students in the experimental group (M = 4.61, SD = 2.149) exhibited higher levels of cognitive engagement compared to students in the control group (M = 4.26 SD = 3.960) , with a large effect size of 0.104. This result suggests that students in the experimental group demonstrated significantly higher cognitive engagement after participating in the BSG learning activities, thereby supporting research question 3.
- Line415-426: The discussion effectively reconnects the results to the study's objectives, highlighting the impact of BSG on engagement and higher-order thinking skills. Further elaborating on how these findings contribute to the broader educational context would be beneficial. This section appropriately interprets key findings, like the impact of BSG on cognitive and behavioral engagement. Incorporating comparisons with existing literature (e.g., past studies on effectiveness) would add depth
Response: Thank you for your suggestion. We added some sentences to incorporating comparisons with existing literature as shown in red words below.
The results s had a beneficial impact on student engagement and higher-order thinking skills and compared to previous studies, which is in line with the previous findings [24,49,50]. In addition, the results show that involving BSG creates learning of skills by simulating real-time experiences in the virtual environment, which is consistent with earlier findings [24,75] and also in line with a study conducted by Deterding et al. [76]
- Line 441-444: While you discuss the implications for CBEC courses, elaborating on broader implications for educational strategies in similar contexts would be enlightening.
Response: Thank you for valuable suggestions to improve the quality four manuscript. We have deleted original sentence and add implication on education s with broader implications as extensive application within business education.
Several studies [24,49,50,51] have illustrated the effectiveness of BSGs in enhancing theoretical underpinnings and establishing a conducive learning environment to facilitate students in cultivating higher-order thinking skills amidst complex challenges. Business simulation games find extensive application within business education, serving as facilitators for students to augment their involvement, elevate higher-order cognitive skills, and attain educational objectives. It is important to emphasize BSGs support education must be implemented. The aforementioned interventions, the promotion of the fourth Sustainable Development Goal (SDG 4) of the 2030 Agenda can be achieved. This objective underscores the importance of providing access to high-quality education for all students and fostering opportunities for lifelong sustainable learning. [77].This approach covers various aspects of the business world, such as understanding market environments and conducting data analysis on e-commerce platforms, along with specific operational and decision-making tasks. Participating in such exercises can improve learners’ systematic thinking and decision-making abilities while enhancing problem-solving, critical thinking, and creativity.
- Line 447-448: The role of teachers in guiding BSG learning is discussed. Expanding on how this facilitation specifically enhances higher-order thinking skills would enrich the discussion.
Response: Thank you for valuable suggestions. We have made supplemented extra data to enrich the discussion with red words.
Thus, teachers should emphasize enhancing students' behavioral engagement, achievable through class discussions, collaborative group work, and various other activities. Consequently, the approach can elevate the overall level of learning engagement, facilitating the improved development of students' higher-order thinking skills, which is in line with the previous findings [50,68].
- Line 476-478: Your suggestion for more research on BSG courses is apt. Including a brief statement on the practical implications for educators and how they might implement these findings in their teaching practice would make the conclusion more impactful.
Consider adding a final sentence or two that broadly encapsulate the study's significance or its implications for the field of veterinary education. This can leave readers with a clear understanding of the study's overall importance.
Response: Thank you for valuable suggestions to improve the quality of our manuscript.
We have added sentences to encapsulate the study's significance or its implications for the field of business education.
It is concluded that educational interventions such as BSGs are crucial innovative tools, enabling educators to design a more engaging curriculum and learning activities that promote both student engagement and higher-order thinking skills. Thanks to technological advancements fostering students' holistic development and delivering high-quality education, in accordance with the 4th Sustainable Development Goal (SDG) since the 2030 Agenda [77] emphasizes achieving a significantly more engaging future education using novel tools and the importance of quality education and skills development [77,78,79].
We tried our best to improve the manuscript and made some changes marked in red in revised manuscript which will not influence the content and framework of the paper. We appreciate for Editors/Reviewers’ warm work earnestly, and hope the correction will meet with approval. Once again, thank you very much for your comments and suggestions.

Reviewer 2 Report
Comments and Suggestions for Authors
The manuscript is interesting, well-written and certainly reports a well-conducted and analysed investigation.
Generally speaking, it is a little difficult for those who are not totally in the scientific field (economics) to understand how business simulation games or CEMO Simulation are structured. Probably a few more details would help the reader less familiar with the subject and broaden the audience of interested persons.
As for the writing, which is very clear in general, there are some typos (see -) and errors in the references (e.g. page 9 ref [66] is [65]).
My only doubt concerns the coherence between the work and its contextualisation in the field of sustainability. I would ask the authors to improve the discussion of this topic to make this link more explicit. If this is not evident, it is unclear why this paper is published in a journal with this title as relevant to the sub-topic Sustainable Education and Approaches.
Then suggest making it more evident in the data analysis where data is compared between groups (see e.g. caption Table4)
Author Response
We feel great thanks for reviewer’s work in our article. According reviewer comments, we response with in italics font as below. Meanwhile, we changes/ additions listed below and the manuscript changed are also given in the red text.
The manuscript is interesting, well-written and certainly reports a well-conducted and analyzed investigation.
Generally speaking, it is a little difficult for those who are not totally in the scientific field (economics) to understand how business simulation games or CEMO Simulation are structured. Probably a few more details would help the reader less familiar with the subject and broaden the audience of interested persons.
As for the writing, which is very clear in general, there are some typos (see -) and errors in the references (e.g. page 9 ref [66] is [65]).
My only doubt concerns the coherence between the work and its contextualization in the field of sustainability. I would ask the authors to improve the discussion of this topic to make this link more explicit. If this is not evident, it is unclear why this paper is published in a journal with this title as relevant to the sub-topic Sustainable Education and Approaches.
Then suggest making it more evident in the data analysis where data is compared between groups (see e.g. caption Table4)
Response:
Thank you for your thoughtful review and valuable feedback on our manuscript. We appreciate your insight into the potential difficulty for readers outside the scientific field, especially in economics, to fully comprehend the structure of business simulation games or CEMO Simulation. We have a brief introduction in 3.3 as Business Simulation Games (BSG) with brief description of each CEMO Simulation section.
In addition, thank you for your insightful review and for highlighting the need for a more explicit connection between our work and its contextualization in the field of sustainability, particularly in relation to Sustainable Education and Approaches. We have modified revise the abstract, discussion(Line 478-482) and conclusion section(Line 522-529) as shown in red sentence to better articulate how our work aligns with the broader context of sustainability with a more comprehensive understanding of the sustainability significance within the specified journal.
As to the writing, we are sorry for our carelessness and have made the corrections “without-“ and change ref [66] to [65] with Line 343 within the whole manuscript. Thanks for your careful checks. Finally, thank you offer the valuable suggestion with data is compared between groups on Table 4. We have added sentence in red to make the it more evident and also modified in the mana manuscript in Line 392 and Line 413.
Abstract:
The 4th Sustainable Development Goal (SDG) of the 2030 Agenda is supported by new educational trends that consider game-based learning as a pedagogical method in the classroom.
Line 478-482
The aforementioned interventions, the promotion of the fourth Sustainable Development Goal (SDG 4) of the 2030 Agenda can be achieved. This objective underscores the importance of providing access to high-quality education for all students and fostering opportunities for lifelong sustainable learning. [77].
Line 522-529
6.Conclusion
It is concluded that educational interventions such as BSGs are crucial innovative tools, enabling educators to design a more engaging curriculum and learning activities that promote both student engagement and higher-order thinking skills. Thanks to technological advancements fostering students' holistic development and delivering high-quality education, in accordance with the 4th Sustainable Development Goal (SDG) since the 2030 Agenda [77] emphasizes achieving a significantly more engaging future education using novel tools and the importance of quality education and skills development [77,78,79]
Line 392
In terms of behavioral engagement, the ANCOVA results in Table 4 show significant differences between the groups (F = 5.274, p < 0.05, 0.025). Based on the post-test mean scores (see Table 3), students in the experimental group (M = 4.69, SD = 1.478) showed higher levels of behavioral engagement compared to students in the control group (M = 4.43, SD = 2.520). According to Cohen[80], an effect size of 0.085 is considered significant.
Line 413
4.1.3 Analysis of Cognitive Engagement
The ANCOVA results in Table 4 show significant differences in cognitive engagement between the groups (F = 6.581, p < 0.05, 0.013). As reflected in the post-test mean scores shown in Table3, students in the experimental group (M = 4.61, SD = 2.149) exhibited higher levels of cognitive engagement compared to students in the control group (M = 4.26 SD = 3.960) , with a large effect size of 0.104.This result suggests that students in the experimental group demonstrated significantly higher cognitive engagement after participating in the BSG learning activities, thereby supporting research question 3.
We tried our best to improve the manuscript and made some changes marked in red in revised paper which will not influence the content and framework of the paper. We appreciate for reviewers’ warm work earnestly, and hope the correction will meet with approval. Once again, thank you very much for your comments and suggestions.

Reviewer 3 Report
Comments and Suggestions for Authors
The conceptual framework is correct, and we agree with the concepts selected for the investigation. The methodologies of flipped learning, problem-solving, project-based learning [PBL], gamification and simulation-games are relevant when pursuing objectives such as student engagement, collaboration, higher-order thinking skills, and creativity. An important variable is the student's satisfaction with their own learning and autonomy in work and decision making.
The problem is the complexity of variables to evaluate the potential of a methodology and in this case more so the use of combined methodologies. That is why the title of the article seems quite ambitious to us.
We want to point out that other influential factors are the training of the teachers who use these methodologies and, no less important, the pre-requisites of the students in the face of the challenge of these methodologies. Additionally, the duration of the learning time is also important. High-level learning objectives take time; learning is not made up of pills but rather long-term sustainability, especially if what we intend is to improve the employability of our students by “equipping them with practical, real-world skills” as the author says.
The references are current, very focused on the research topic and very well selected. We can say that the author has well founded his research.
We are going to comment on some aspects hoping to help the author in his future investigations:
1. Regarding research questions, we would advise fewer questions or establishing a relationship between them.
2. Behavioral engagement is related to abilities, capacities and skills of action and resolution, for which a cognitive base is needed; They form the interrelation of theory and practice, while emotional engagement is essential for the practice to be successful and the learner to be satisfied with their learning. They are completely interrelated.
3. On the other hand, High-order Thinking Skills (HOTS) is the highest level of cognition in practical skills, it is not necessary to consider each element separately since the learner is an integral person and learning must be integral, not a sum of parts.
4. The potentials of flipped learning and gamification are based on the ability to offer this interrelation of learning that enables subsequent decision-making.
5. A problem is the heterogeneity of prerequisites in students, which affects the evaluation of the work.
6. Therefore, we must look for a methodology that adapts to their differences and an evaluation model that establishes the difference between the starting point of each student in relation to her achievements.
7. Qualitative interviews are essential to understand the difficulties of the work, assess satisfaction with the work done and with the help received, therefore they should have been better used in the research.
We would have liked to know the items of the texts used since HOTS consume the three types of engagements. Problem-solving is a practical and cognitive skill at the same time; it is impossible to do it without a conceptual framework. And the same thing happens with Critical thinking, it is a cognitive capacity that, if it does not act on practical knowledge, will not be effectively critical. You cannot think or be creative about nothing, there must be practical manipulation. For this reason, we affirm that it cannot be denied that good critical thinking demands creativity and to solve a problem we need creativity and critical thinking. The opposite is pure reproductive memory.
We agree with the author that “Incorporating BSG into this context creates a learner-centered environment that can significantly improve students…” … Deep learning: it is not necessary to isolate each element of good learning.
“The apparent lack of effect on emotional engagement…” in our opinion may be a failure of the type of test used, because it is impossible for a student to have carried out a good project if the methodology and the teaching staff have not managed to involve it. Perhaps the question would be simpler, is the apprentice satisfied with his effort and the achievements obtained.
We believe that the research deserves to be published and we hope that our comments can help you.

Author Response
Thank you for your detailed and constructive feedback on our manuscript. We appreciate your acknowledgment of the correctness of the conceptual framework and the relevance of the selected concepts for investigation. We understand your concerns about the ambitious nature of our article's title, particularly in the context of evaluating the potential of combined methodologies. We genuinely appreciate your guidance and will carefully consider all your suggestions for future investigations. Your feedback is instrumental in enhancing the quality and impact of our work.
Thank you once again for your time and thoughtful review.

Round 2
Reviewer 1 Report
Comments and Suggestions for Authors
Dear Authors,
Comment 1
Thank you for the revisions made in the section discussing the effectiveness of flipped classrooms. Your detailed overview of the flipped classroom methodology, its application across various fields, and its alignment with Bloom's Taxonomy is appreciated. It adds valuable context to your study.
However, I note that the specific study I recommended in my previous comment (https://doi.org/10.1016/j.jevs.2023.104537) has not been included. This study provides crucial insights into student perceptions of flipped classrooms, which could significantly enrich your discussion. Including such recent and relevant research is critical for grounding your arguments in the latest findings and enhancing the credibility of your work.
Additionally, the initial lines 90-91 of your revised section, while informative, do not appear to be directly supported by cited research. It is crucial for assertions, especially those concerning the increasing use and effectiveness of flipped classrooms, to be based by empirical evidence.
Comment 2
Thank you for expanding the section on Project-Based Learning (PBL) in lines 148-164. While this addition provides a good overview of PBL, it could benefit from further enhancement to match the detail in the sections on flipped classrooms and game-based learning:
Lines 158-162: Please add detailed examples or case studies to illustrate PBL in action, especially in business education. This will give readers a clearer picture of its practical application and effectiveness.
Lines 150-154, 158-162: Strengthen this section by citing specific studies or data demonstrating PBL's impact on student outcomes in higher education.
Author Response
Response:
Thank you for your feedback. However, based on the reviewer comments from round 1, there was no mention of the flipped classroom and specific study at (https://doi.org/10.1016/j.jevs.2023.104537). In this time, we’ve incorporated the study you provided (https://doi.org/10.1016/j.jevs.2023.104537) as reference of no.81 into this article to enhance the depth of research.
The flipped classroom describes a mixed methodology that combines face-to-face and virtual teaching methods and is currently being employed at all educational levels. Its increasing use has been attributed to its overall effectiveness and the specific practical components that constitute its training approach [22, 23, 81].
Response:
Thank you for your valuable suggestions. We have made supplemented extra case studies to illustrate PBL in business education and specific studies demonstrating PBL's impact on student outcomes in higher education in red words.
# Lines 150-154
The research [82] implemented PBL in a business informatics university course. In this action, students acquire knowledge through practical application during the project elaboration, aligning with the principles of PBL. Throughout the process, students participate in hands-on activities, including exploring the basics of data processing, conducting data analysis, modeling business processes, and developing a simple system. Additionally, this pedagogical approach proves to be a highly effective method that seamlessly integrates into dynamic and demanding learning environments such as international business education [74]. In higher education, PBL enables students to gain a broad spectrum of knowledge and essential innovative skills crucial for addressing future challenges and attaining success [83]. Several studies have highlighted the positive influence of PBL on students' attitudes toward learning, resulting in enhanced effectiveness and engagement [82,83,84,85].
# Lines 158-162:
In higher education, PBL learning based learning equips students with a varied spectrum of knowledge and essential innovative skills, enabling them to effectively navigate future challenges and achieve success [71]. PBL has the potential to positively influence students' attitudes toward learning, leading to increased positive effects on both student learning effectiveness and engagement. [82-85].
